# Use of Poly Lactic-co-glycolic Acid Nano and Micro Particles in the Delivery of Drugs Modulating Different Phases of Inflammation

**DOI:** 10.3390/pharmaceutics15061772

**Published:** 2023-06-20

**Authors:** Chiara Puricelli, Casimiro Luca Gigliotti, Ian Stoppa, Sara Sacchetti, Deepika Pantham, Anna Scomparin, Roberta Rolla, Stefania Pizzimenti, Umberto Dianzani, Elena Boggio, Salvatore Sutti

**Affiliations:** 1Department of Health Sciences, Università del Piemonte Orientale, Via Solaroli 17, 28100 Novara, Italy; 20032501@studenti.uniupo.it (C.P.); luca.gigliotti@med.uniupo.it (C.L.G.); ian.stoppa@uniupo.it (I.S.); sarasacchetti1996@gmail.com (S.S.); deepika.pantham@uniupo.it (D.P.); roberta.rolla@med.uniupo.it (R.R.); elena.boggio@med.uniupo.it (E.B.); salvatore.sutti@med.uniupo.it (S.S.); 2Maggiore della Carità University Hospital, Corso Mazzini 18, 28100 Novara, Italy; 3NOVAICOS s.r.l.s, Via Amico Canobio 4/6, 28100 Novara, Italy; 4Department of Drug Science and Technology, University of Torino, 10125 Turin, Italy; anna.scomparin@unito.it; 5Department of Physiology and Pharmacology, Sackler Faculty of Medicine, Tel Aviv University, Tel Aviv 69978, Israel; 6Department of Clinical and Biological Science, University of Turin, Corso Raffaello 30, 10125 Torino, Italy; stefania.pizzimenti@unito.it

**Keywords:** PLGA nanoparticles, oxidative stress, IBD, cardiovascular diseases, transplant rejection, neurologic diseases, osteoarthritis, bone regeneration, skin wound healing, eye diseases

## Abstract

Chronic inflammation contributes to the pathogenesis of many diseases, including apparently unrelated conditions such as metabolic disorders, cardiovascular diseases, neurodegenerative diseases, osteoporosis, and tumors, but the use of conventional anti-inflammatory drugs to treat these diseases is generally not very effective given their adverse effects. In addition, some alternative anti-inflammatory medications, such as many natural compounds, have scarce solubility and stability, which are associated with low bioavailability. Therefore, encapsulation within nanoparticles (NPs) may represent an effective strategy to enhance the pharmacological properties of these bioactive molecules, and poly lactic-co-glycolic acid (PLGA) NPs have been widely used because of their high biocompatibility and biodegradability and possibility to finely tune erosion time, hydrophilic/hydrophobic nature, and mechanical properties by acting on the polymer’s composition and preparation technique. Many studies have been focused on the use of PLGA-NPs to deliver immunosuppressive treatments for autoimmune and allergic diseases or to elicit protective immune responses, such as in vaccination and cancer immunotherapy. By contrast, this review is focused on the use of PLGA NPs in preclinical in vivo models of other diseases in which a key role is played by chronic inflammation or unbalance between the protective and reparative phases of inflammation, with a particular focus on intestinal bowel disease; cardiovascular, neurodegenerative, osteoarticular, and ocular diseases; and wound healing.

## 1. Introduction

Inflammation is an evolutionarily conserved process that defends the host against pathogens and plays a fundamental role in mediating tissue healing after injury [1]. It evolves in a first defensive/aggressive phase aimed to counteract infections and remove necrotic tissues and a second reparative phase aimed to regenerate the damaged tissue or, in the case of extended damages, to substitute it with fibrotic tissues. However, persistence or dysregulation of resolving mechanisms of the acute inflammatory response may cause cellular injury per se, resulting in chronic inflammation that destroys tissue architectures and contributes to organ failure [2]. In addition, locally unresolved or uncontrolled inflammation produces large amounts of molecular mediators that, once released into the bloodstream, may contribute to the spread of inflammation systemically [3]. Chronic inflammation is a key mechanism contributing to the pathogenesis and progression of many diseases [2,4], and its systemic involvement supports the development of apparently unrelated pathological conditions such as metabolic disorders, cardiovascular diseases, and tumors [3]. Therefore, targeting inflammation represents a wide-spectrum strategy to contrast the evolution of several diseases toward advanced stages [5]. Despite that, the use of anti-inflammatory compounds has its drawbacks [6]. Indeed, long-term treatment with conventional anti-inflammatory medications such as corticosteroids and nonsteroidal anti-inflammatory drugs (NSAIDs) may cause severe adverse effects, including cardiovascular complications, diabetes, gastrointestinal ulcers, and bleeding [7,8,9,10,11]. In addition, some medications show a short half-life, thus requiring multiple administrations to maintain therapeutic concentrations [7]. To overcome some of these limitations, researchers pursued alternative strategies, including targeted delivery of traditional anti-inflammatory molecules to reduce their off-target effects and the use of novel natural bioactive compounds displaying anti-inflammatory properties with limited side effects [12]. However, the clinical application of natural compounds is often challenging owing to their scarce solubility and stability associated with low bioavailability [12]. In this regard, recent studies showed that encapsulation within nanoparticles (NPs) may represent an effective strategy to enhance the pharmacological properties of bioactive molecules [13]. Moreover, NPs can be functionalized for targeted therapy, avoiding the intrinsic adverse effects associated with anti-inflammatory treatments [7].

Amongst NPs, poly lactic-co-glycolic acid (PLGA)-NPs have found an extensive spectrum of applications since they are one of the most versatile drug delivery systems [14]. PLGA-NPs offer multiple advantages in enhancing the therapeutic efficacy of anti-inflammatory drugs, improving bioavailability and biodistribution, and ensuring controlled release [15,16]. PLGA is an FDA-approved, biodegradable, and biocompatible synthetic polymer widely used to develop nano-sized drug delivery carriers [17]. PLGA features, such as erosion time, hydrophilic/hydrophobic nature, and mechanical properties, can be controlled by tuning the polymer’s molecular weight and the lactide/glycolide molar ratio. Furthermore, the PLGA backbone presents reactive groups available for the conjugation of targeting molecules [18], antibodies [19], or imaging probes [20]. PLGA can be easily conjugated to other polymers to obtain co-polymers with enhanced pharmaceutical properties [21], able to form NPs, micelles, or nanogels [22]. The physicochemical and biological behavior of the PLGA-based delivery systems also depends on the preparation technique. In particular, the emulsification and evaporation [23] or the nanoprecipitation techniques [24] are useful for encapsulating hydrophobic molecules. Nevertheless, other techniques, such as solvent diffusion [25] or phase-inversion [26], can be used to avoid exposure of the active molecules to harsh conditions (high energy, a potential increase of temperature, etc.) of the emulsification process. Typically, the size of PLGA NPs is in the range of 10–1000 nm, and those below 200 nm can extravasate from leaky blood vessels, accumulate, and penetrate tissues via the enhanced permeation and retention (EPR) effect [27]. Due to this capacity, they have been widely exploited to deliver anticancer drugs to tumors [28,29]. In addition, they find broad use in the treatment of cardiovascular disease [30], immunotherapy [31], including several types of vaccines [32,33,34,35], and tissue regeneration [36].

PLGA-NPs represent an interesting delivery system to treat inflammatory diseases as well [37]. As mentioned above, PLGA is biodegradable, and one of its degradation products, lactic acid, has an intrinsic immunosuppressive, anti-inflammatory effect [38]. Furthermore, PLGA-NPs, like many other colloidal nanocarriers, are easily taken up by the immune system, allowing for the selective delivery of anti-inflammatory drugs [39]. In general, NPs between 10 and 200 nm are commonly used as drug delivery systems, with size-dependent toxicity (smaller particles are more toxic). NPs with an average size below 200 nm are typically internalized by dendritic cells, while NPs above 500 nm are preferentially captured by macrophages [40]. This specific interaction with immune cells depends primarily on the size of the NPs, but this is not the only factor involved. Also, surface properties, such as charge, hydrophobicity, and chemical modification, deeply affect the fate of the NPs [41]. The conjugation of targeting agents on the surface of NPs for the specific recognition of the surface of target cells is a well-established strategy to improve the cell-specific uptake of the nanomedicines. The most commonly used targeting agents are antibodies and low molecular weight ligands for cellular receptors. The specific molecules are selected according to the physio-pathological characteristics of the target cells [42]. For instance, glycosaminoglycans, such as hyaluronic acid and chondroitin sulfate, that bind selectively to CD44 receptors overexpressed by macrophages are commonly used in inflammatory bowel diseases [43,44]. Alternatively, ICAM1 and MAdCAM1, ligands of α4β7 integrin, were shown to increase the accumulation of NPs in a murine model of acute colitis [45]. Integrins are a good target to increase the site-specific accumulation of NPs at the inflammation site. A cyclic RGD peptide (arginine-glycine-aspartic acid) targeting αvβ3 integrins expressed on angiogenic vascular endothelial cells improved the accumulation of NPs in a Rheumatoid Arthritis model [46].

PLGA-NPs are particularly promising carriers to treat inflammatory diseases due to the intrinsic features of the polymer, such as biocompatibility and biodegradability. Despite their hydrolyzable nature, they have an excellent shelf-life and can improve the stability of drugs in physiological fluids [47]. The encapsulation efficiency (EE%) for this type of formulation depends largely on the physicochemical properties of the drug and can be modulated with chemical modification of the polymeric structure or different preparation techniques [23,24].

The main drawbacks of this type of formulation are drug loading (typically around 1% *w*/*w*) and the high burst release of the loaded cargo. Furthermore, high production costs and difficulties with large-scale production can limit the PLGA-NPs’ applicability in clinical practice [48]. Indeed, great efforts have been made at the industrial level to promote scale-up and improve PLGA-NPs manufacturing [49].

PLGA-NPS are not the only colloidal systems exploited for the treatment of inflammatory diseases. Lipid NPs [50], gold NPs [51], and exosomes [52] are common alternatives to polymeric systems. Each nano-sized system exploits its own features to achieve an enhanced anti-inflammatory effect. Among them, exosomes are particularly interesting since they have an intrinsic immunomodulatory activity, depending on the cells from which they originate. The number of studies involving exosomes is rising; however, for this technology, manufacturing issues are a limiting factor to clinical applicability [52].

In this review, we focus on PLGA-based NPs, as this commercially available polymer has been used in FDA-approved medicines for more than 30 years and can be produced at a GMP-grade standard for clinical applications [53], making it an excellent candidate for the clinical translation of NPs.

A huge bulk of studies are focused on the use of PLGA-NPs to deliver immunosuppressive treatments for autoimmune and allergic diseases or to elicit protective immune responses, such as in vaccination against infectious agents and cancer immunotherapy, and have been recently reviewed and are summarized in Table 1 [31]. By contrast, this review is focused on the use of PLGA NPs, often with a topical approach, in preclinical in vivo models of other diseases in which a key role is played by chronic inflammation or unbalance between the protective and reparative phases of inflammation.

## 2. Oxidative Stress

Inflammation is intimately connected to oxidative stress since inflammatory cells, such as phagocytes, liberate several reactive oxygen or nitrogen species (ROS/RNS) at the site of inflammation, which worsen the inflammatory status and activate signaling pathways, such as the NF-κB signal, inducing the expression of various pro-inflammatory genes [54,55,56,57].

ROS/RNS, which include both radical and non-radical species [58], are constantly produced in living organisms. During the normal cellular metabolic processes, cells can produce ROS such as hydrogen peroxide (H_2_O_2_), superoxide radical (^•^O_2_^−^), and RNS such as nitric oxide (^•^NO), involving both nonenzymatic and enzymatic (i.e., adenine dinucleotide phosphate (NADPH) oxidases (NOXs), myeloperoxidase (MPO)) pathways [58,59]. ROS/RNS can act as intracellular signaling molecules, regulating physiological cellular functions and the homeostatic adaptation of cells to external stimuli, and they can be overproduced during infection since they are highly toxic to pathogens [60].

However, when in the condition of oxidative stress, ROS/RNS are produced in excess, becoming harmful compounds. At high concentrations, they can damage cellular DNA, proteins, and lipids. In lipid membranes, ROS/RNS induce lipid peroxidation, with the formation of highly reactive aldehydes, including malondialdehyde (MDA), further amplifying the toxic effect of free radicals [61].

Antioxidant enzymes, such as superoxide dismutases (SODs), catalase (CAT), or glutathione peroxidases (GPXs), and nonenzymatic antioxidant endogenous molecules, like glutathione (GSH), are involved in free radical detoxification [62]. The expression of a wide spectrum of endogenous antioxidants is regulated by nuclear factor erythroid 2-like (Nrf2 or NFE2L2), considered the master regulator of the cytoprotective response upon binding to the antioxidant response element (ARE) sequences present in the promoter of antioxidant genes, such as those involved in GSH synthesis [63].

Many types of PLGA-based nano- or microsized particles and hydrogels have been developed to contrast oxidative stress by delivering antioxidant drugs or inhibiting excessive inflammation, obtaining a concomitant lowering of excess ROS/RNS levels. Of note, both oxidative stress and inflammation are important parameters in the hazard assessment of polymeric nanomaterials, including PLGA NPs [64].

Al-Shalabi and collaborators investigated the antioxidant and anti-inflammatory abilities of the bioactive plant flavonoid Rhoifolin (ROF) loaded in PLGA NPs (ROF NPs) in formalin-induced rat paw edema, a model of chemical-induced inflammation [12]. They showed that ROF NPs are more active than free ROF in inhibiting the thickness of the paw edema and mitigating the histopathological changes in the inflamed paw tissues (assessed in terms of lymphocyte, neutrophil, and mast cells infiltration) and vasculitis. ROF NPs were able to normalize the gene expression profile of pro-inflammatory cytokines (TNF-α and IL-1β) and to down-regulate the level of ^•^NO, a well-known free radical promoting vasodilatation during inflammation [65]. Additionally, they demonstrated antioxidant properties in vitro, showing that ROF NPs are more effective than free ROF in preventing the ROS increase induced by with tert-Butyl hydroperoxide (*t*BHP) in RAW 264.7 macrophages [12].

Chemical-induced inflammation was also obtained in a model of an ethanol-induced gastric ulcer by Chakraborty et al., who investigated the protective role of the bioflavonoid quercetin (QC) loaded in PLGA NP. In particular, they found that QC PLGA NPs are 20-fold more efficacious than free QC in preventing gastric ulcers by inhibiting inflammatory cell infiltration and oxidative damage in rat gastric tissues, measured as a decrease of lipid peroxidation level, MPO activity, and ROS content, as well as an increase of the antioxidant GSH [66].

Pulmonary inflammation can be achieved by exposing animals to bacterial lipopolysaccharide (LPS). Kotta and collaborators [67] investigated the anti-inflammatory activity of dexamethasone (DEXA), a synthetic corticosteroid, loaded in lipopolymeric microspheres of PLGA in an LPS-induced Acute Respiratory Distress Syndrome (ARDS) animal model. Intravenous injection of DEXA microspheres successfully down-regulated inflammation markers, such as IL-1β, IL-6, and TNF-α, in the bronchoalveolar lavage fluid. Moreover, the levels of lipid peroxidation were inhibited, suggesting a reduction of oxidative stress [67].

Wrapping PLGA NPs with red blood cell (RBC) membranes is an emergent strategy to lengthen the time of circulation and to better avoid the immune recognition of NPs. This type of nanovehicle was successfully used to deliver basic Fibroblast Growth Factor (bFGF), a member of the fibroblast growth factor family with antioxidant and anti-inflammatory properties, in a sepsis-induced cardiac injury in vivo model [68]. In heart tissues, both oxidative stress (GSH, GPX, CAT, SOD, MPO, and MDA) and inflammatory (IL-1, IL-6, TNF-α, and HMGB1) markers were determined, demonstrating that bFGF loaded in RBC membrane-coated PLGA NPs had therapeutic benefits in the treatment of the cardiac injury.

The possibility to synthesize PLGA-NPs tailored to target specific inflammatory cells such as neutrophils and macrophages, key players in supporting oxidative stress, has allowed for their use in modulating acute and chronic inflammatory disorders [37,69,70,71,72].

## 3. Inflammatory Bowel Disease

Inflammatory bowel disease (IBD) is a chronic-relapsing inflammatory disorder of the gastrointestinal (GI) tract, whose two major forms are Crohn’s disease (CD) and ulcerative colitis (UC), and its incidence is progressively rising worldwide [73]. While UC is usually confined to the large bowel, especially the rectus and anus, CD affects the entire GI tract, from the mouth to the anus, but most commonly the terminal ileum and the perianal region [74]. From a histopathological point of view, both CD and UC share a disrupted intestinal mucosa, with increased permeability and alteration of mucus thickness and composition [75], but while CD is characterized by a predominantly transmural inflammatory cellular infiltration, UC inflammation invades only the superficial intestinal lining [76,77]. Despite some differences in terms of risk factors, pathophysiology, and clinical manifestations, both diseases share a dysregulated immune response, so they can be considered immune-mediated disorders. Indeed, their pathogenesis seems to derive from a combination of genetic predisposition (for instance, polymorphisms in genes involved in the inflammasome pathway), environmental modifiers of intestinal immunity (such as diet, smoking, antibiotics, and vitamin D status), and a disrupted immune response to commensal gut microbiota [78,79], with an ecological imbalance between beneficial and harmful symbiotic bacteria and an excessive bacterial translocation through the intestinal barrier [80].

Owing to the predominantly immune-mediated pathogenesis, IBD treatment is focused on drugs aimed at suppressing the inflammatory process. The current therapeutic strategies include 5-aminosalicylic acid (5-ASA), immunosuppressants, such as methotrexate, cyclosporine, or tacrolimus (TACR), monoclonal antibodies directed against key pro-inflammatory mediators such as TNF-α, IL-12, and IL-23, and small molecules able to interfere with the inflammatory cascade, such as the Janus kinase inhibitors tofacitinib and filgotinib or immunomodulators like ozanimod [81,82]. However, all these treatments share the risk of nonspecific systemic immunosuppression and significant adverse events [83], and, especially when administered parenterally as is the case of anti-TNF-α biologics, they might reach suboptimal levels at the inflamed bowel sites [84]. Furthermore, most anti-inflammatory drugs are absorbed already in the small intestine, leading to systemic diffusion and little effect on the distal intestinal segments where they are supposed to act. Degradation by the gastric acidic pH or small intestinal digestive enzymes is another factor jeopardizing the safe transport to the low GI [85]. The real challenge would be a therapy specifically targeting the GI tract. Unfortunately, the development of prodrugs that lose their hydrophilic portion only after being processed by the bacterial metabolism or release their active part in the large bowel owing to the favorable biochemical environment has only partially overcome this problem since the main caveat is their scarce sustained release, which necessitates an increase in the dosage and the frequency of administration with an impact on patient’s compliance [86,87,88]. Moreover, a frequent manifestation of IBD is diarrhea, which impairs the drug’s ability to act locally for a prolonged time, especially when using macroscopic drug delivery systems [89,90].

A possible solution is the use of NPs as drug-delivery vehicles since targeted delivery and controlled release are two key features of these pioneering biomaterials. In particular, PLGA NPs possess ideal features since they are biocompatible with little effect on cell viability [85,91], are easily deposited in inflamed tissues, and are sufficiently biodegradable to prevent polymer accumulation [91]. Furthermore, they show little inflammatory potential and immunogenicity [92].

Owing to the inflammation-related pathogenesis of IBD, it is not surprising that an approach to its treatment could precisely exploit the inflammatory component to improve drug delivery. The inflammatory reaction of IBD involves predominantly cell-mediated immunity, where macrophages, neutrophils, natural killer (NK) cells, and T cells infiltrate the gut walls and are responsible for ongoing and self-amplifying inflammation. One of the advantages of NPs is their ability to be taken up by these inflammatory cells due to their small size. In this way, not only can they target precisely the cells responsible for IBD pathogenesis, increasing the selectivity of the therapy, but they can also exert a long-lasting effect, being trapped inside them. Their polymeric composition also allows them to adhere to the mucus produced by the gut mucosa already in physiological condition and especially during inflammation, where a thicker mucus layer accumulates in the inflamed areas [86]. In other words, the inflammation underlying IBD becomes the expedient through which PLGA particles penetrate the gut mucosa more efficiently. Their distinct pathway of absorption and delivery may be summarized with a combination of enhanced permeability, increased mucus production, and greater infiltration of inflammatory cells [90,92] (Figure 1).

PLGA particles prepared with the sonication or ultracentrifugation method have a spherical, smooth, and pore-free surface and a submicrometer size [85,86,90,93,94]. An essential issue to consider is the optimal particle size, which should reflect a compromise between the drug-loading capacity, which is surely higher in larger particles, and the delivery efficacy that is inversely related to the size. In general, the smaller the particle, the greater the accumulation, with the best size ranging from 100 to 500 nm [86,94,95]. Particle size can also be exploited to differentially target the intestinal barrier. To prove this concept, Mohan and collaborators worked on human colorectal adenocarcinoma (Caco-2) cell lines grown in culture and triggered with pro-inflammatory cytokines to mimic inflammation, showing that small particles (around 100 nm in diameter) can target both the epithelial barrier and the basolateral side of enteric cells, while middle-sized particles in the 200–300 nm range are more suited for basolateral accumulation and large ones (around 600 nm in diameter) limit their action to the apical cell membrane. The differential bioengineering of PLGA particles represents thus a potential strategy to target the inflammatory cells of the basolateral *lamina propria* [93].

The processing technique is also a key factor for optimal drug encapsulation. For instance, emulsification proved to be more efficient than solvent diffusion because, with the latter one, the drug tends to be extracted and lost towards the aqueous phase [92,96]. When an encapsulation-free system is used, particle size is again a key factor since the available surface area, which is inversely correlated with the particle dimension, impacts the molecule adsorption [97].

The kinetics of drug release from the NP involves a first burst within the first 30 min, likely due to the dissolution of the drug adsorbed on the particle surface, followed by a more sustained release of the encapsulated molecules lasting about 24 h [91,95]. To improve the drug-loading efficacy, the preparation may involve some washing steps to minimize the drug adsorption on the particle surface, resulting in an improved encapsulation of the molecule and a sustained release to the target site [95]. The proposed mechanism of drug release is simple diffusion out of the NP, at least in vitro, but it should be considered that, in vivo, a more complex scenario may take place, with possible enzymatic degradation of the nanoparticle and a resulting faster release of the encapsulated molecule [95]. Interestingly, the degradation time also depends on the particle molecular weight and the lactic-to-glycolic acid ratio, which could be modified to one’s advantage during the manufacturing process [96].

In 2001, Lamprecht and collaborators compared the administration of free versus PLGA-encapsulated rolipram (ROL), an anti-depressant drug with proven anti-inflammatory activity thanks to its anti-phosphodiesterase effect, in an IBD rat model. The typical strong first-pass metabolism of ROL was significantly reduced when it was administered inside PLGA NPs. Moreover, although the anti-inflammatory effects were similar in the two treatments, mice receiving PLGA-encapsulated ROL experienced significantly fewer relapses compared with those treated with a free ROL solution, which suggests a sustained local release of the drug [86].

A similar approach was used by Tahara et al., who used PLGA particles to deliver a nuclear factor (NF)-κB decoy oligonucleotide to the inflamed gut mucosa of IBD mice to reduce inflammation by interfering with the NF-κB activity promoting inflammation. The encapsulation of a synthetic double-stranded oligonucleotide (ODN) protected the nucleic acid from the harsh GI microenvironment and allowed its targeted delivery to the colon mucosa. The results were a prophylactic effect on bowel shortening and reduced inflammation [85]. However, the safe delivery of the oligonucleotide decoy receptor required an additional precaution during the PLGA particle preparation, i.e., surface modification with the cationic polymer chitosan to improve their mucoadhesiveness. Indeed, naked ODN or ODN loaded in plain PLGA did not show any therapeutic efficacy, stressing the importance of the thoughtful manufacturing of these biomaterials [85]. The reason is that both the gut mucosa and the PLGA NP surface are negatively charged, which is a possible limitation to PLGA particle absorption [98]. However, complexing with chitosan could overcome this obstacle since this polymer opens the tight junctions between epithelial cells [96,99] and favors electrostatic interactions with the negatively charged cell membranes and the negative mucins of the intestinal mucus [99]. Moreover, chitosan is not degraded in the upper GI tract, being enzymatically hydrolyzed by colonic bacteria instead [100]. PLGA particle functionalization with other polymers can be achieved in a similar way using PEG. A study by Shrestha et al. demonstrated that even the PEG chain length affects the in vivo penetrating performance of NPs, likely due to the change in hydrophilicity according to the polymer length [101].

Most of the differential bioengineering techniques in PLGA particle preparation (PLGA-PEG, PLGA-chitosan, and non-functionalized PLGA micro- and nanoparticles) have been recapitulated in one of the few observational studies on human subjects performed by the group of Lautenschläger. Their research reiterated some important concepts. First, inflammation is the entry key for enhanced penetration of both micro- and nanoparticles. Second, independently of the inflammatory state of the mucosa, the delivery efficacy is inversely related to particle size. Third, PLGA particle functionalization has a strong impact on its functional behavior, with PLGA-chitosan particles being more suited for bioadhesion and PEG-functionalized ones being much more adequate for deep penetration and translocation [94].

Dealing with less conventional applications of PLGA particles in IBD therapy, the chitosan strategy has been tried in vitro by Bai et al. to test the efficacy of the anti-inflammatory and immuno-modulatory medicinal fungus *Phellinus igniarius* (PI). PLGA NPs loaded with PI polysaccharide (PIP) and combined with chitosan proved to be superior to naked PIP and PLGA-PIP without chitosan at reducing inflammation of experimentally induced colitis and prevented pathological colon shortening, improved the reparability of the intestinal barrier, and ameliorated microbial diversity to such an extent that the authors dare to define it as a prebiotic [99].

Similarly, the pleiotropic (anti-inflammatory, anti-carcinogenic, antioxidant) effects of the polyphenol curcumin (CURC) have been applied to IBD treatment using PLGA particles. CURC exerts anti-inflammatory effects by interacting with the nuclear receptor peroxisome proliferator-activated receptor gamma (PPAR-γ) [102]. However, its hydrophobic nature hampers its clinical applications because of its intestinal absorption and low bioavailability [103]. Recent studies have demonstrated that entrapping CURC in PLGA-NPs increases its bioavailability leading to effective therapeutic concentrations [104]. This observation paved the way to explore its efficacy in the context of inflammatory disorders. In this regard, Hlaing et al. provided an intriguing example of PLGA-NP-based target therapy delivering CURC directly to macrophages. Briefly, they fine-tuned CURC-encapsulated hyaluronic acid (HA)-conjugated NPs (CURC-HA-PLGA-NPs) to target inflamed colitis tissue, exploiting the HA capacity to bind CD44, a receptor widely expressed on activated macrophages. The authors revealed that CURC-HA-PLGA-NPs specifically target inflammatory macrophages, lowering the production of inflammatory mediators and reverting the histopathological features associated with colitis [105]. Similarly, CURC entrapped within PLGA particles combined with Eudragit^®^ S100, a mixture of pH-sensitive polymethacrylate polymers, displayed increased anti-inflammatory and protective effects in experimental colitis mice compared to free CURC [106].

Finally, an additional intriguing application of PLGA NPs is their use in diagnostic imaging. Since expression of mucosal vascular addressin cell adhesion molecule-1 (MADCAM-1) represents a reliable, specific, and early marker of IBD-related inflammation, Truffi et al. investigated the use of PLGA-PEG micelles conjugated with monoclonal antibodies against MADCAM-1 and loaded with manganese oxide (MnO) nanocrystals as contrast enhancers in magnetic resonance imaging (MRI). These NPs were intravenously injected into IBD mice. The T1-weighted images obtained after 24 h showed enhancement in the rectal mucosa, proving the specific localization of PLGA NPs [107]. This technique mimics the behavior of circulating leukocytes during inflammation and allows for the tracking of their interaction with adhesion molecules whose expression is enhanced during inflammatory conditions. In a clinical scenario, this approach might represent a revolutionary noninvasive imaging tool to be used in the first diagnosis and follow-up, allowing precise mapping of the diseases and reducing the need for repeated biopsies.

## 4. Cardiovascular Diseases

Cardiovascular diseases (CVD) encompass different pathological conditions ranging from atherosclerosis (AS) to myocardial infarction (MI), ischemic stroke, and gangrene [108]. Among them, AS is considered the principal contributor to CVD, being responsible for narrowing the lumen of arteries and inducing atherothrombosis, which hampers the blood supply to vital organs such as the heart [109]. The earliest visible lesion of AS is the fatty streak owing to the accumulation of oxidized lipoprotein-loaded macrophages (also known as foamy cells) that progressively populate the intima layer of medium- and large-sized arteries, sustaining inflammation throughout plaque formation [110]. In the long run, these early lesions induce chronic inflammatory responses and extracellular matrix (ECM) remodeling, thus evolving into fibrous plaques, which may undergo surface erosions, becoming vulnerable [111]. The instability of atherosclerotic plaques leads to ruptures, exposing thrombogenic material to the blood coagulation system that may cause thrombus formation, artery occlusion, and MI [112,113,114]. AS is regarded as a chronic inflammatory disorder in which dyslipidemia, oxidative stress, and macrophage activation play an essential role [115]. In this pathological context, dyslipidemia fosters macrophage activation followed by the recruitment of myeloid cell pools within the artery wall, sustaining inflammation and oxidative stress that, in their turn, promote plaque growth [115]. Therefore, therapeutic strategies targeting such pathogenetic factors may theoretically impact the course of AS, not only by preventing infarcts but also having critical implications on the injury and tissue repair once MI occurs [116,117]. This latter assertion is supported by the fact that macrophages, based on their activation state, display a dual role in infarcted heart tissues, on the one hand mediating the injury and, on the other hand, the healing process [118]. Currently, sizable efforts are underway to counteract the global health burden represented by CVD, searching to optimize the pharmacological use of new and old drugs, trying to neutralize detrimental effects associated with environmental and behavioral factors, or directly targeting plaque-causing cells [119]. PLGA-NPs found a large spectrum of biomedical applications in the treatment of AS and its cardiovascular consequences, such as MI [120,121].

In this regard, to successfully treat atherosclerosis, Liang et al. [122] devised a highly sensitive H_2_O_2_-scavenging PLGA-based nano-bionic system loaded with probucol (PU), a lipid-regulating agent with anti-oxidative and anti-inflammatory effects. The H_2_O_2_-scavenging polymer of PLGA-diphenyl ring-peroxyoxalate bond-PEG (6s-PLGA-DAr-PO-PEG) loaded with PU was coated with erythrocyte membranes (RPP-PU) then intraperitoneally injected in atherosclerosis-prone apolipoprotein E-deficient (Apoe^-/-^) mice. RPP-PU effectively eliminated pathological ROS, reduced the level of lipids, and significantly decreased the area of vascular plaques and fibers. Another approach used PLGA-NPs loaded with colchicine (COLC) and modified to target macrophage function in the context of AS [67]. COLC is a powerful anti-inflammatory alkaloid used since the days of Ancient Egypt to treat inflammatory disorders [123,124]. In recent years, COLC experienced a renaissance for its possible use in dermatology and to prevent AS plaque formation [125,126]. Unfortunately, its clinical use is limited because of the narrow therapeutic index that increases the risk of severe side effects [127]. To overcome these limitations, Li Y et al. synthesized COLC-loaded PLGA-NPs coated with modified cellular membranes overexpressing the integrin VLA-4 (a homing receptor) and CD47 (acting as a do not eat me signal for macrophages) to enhance targeting the atherosclerotic plaque and inhibit phagocytosis, respectively. The efficacy of these specifically modified PLGA-NPs has been tested in vulnerable atherosclerotic plaque mice, observing decreased macrophage infiltration and enhanced plaque stability [67]. A further example of targeted therapy applied to AS has been described by Zhu Y et al., who developed PLGA-PEG NPs carrying a benzothiazole compound (called 5242331) able to upregulate the ATP-binding cassette transporter A1 (ABCA1), a protein expressed on the cell surface of macrophages that stimulates the reverse cholesterol transport from peripheral body districts to the liver by inducing intracellular cholesterol efflux and high-density lipoprotein (HDL) formation. To selectively target atherosclerotic plaque and foam cells, the authors functionalized such PLGA-NPs with the PP1 peptide, which binds with high affinity to the scavenger receptor-AI (SR-AI) expressed by foam cells within atherosclerotic plaques. The in vivo application of these specifically modified PLGA-NPs induced atherosclerotic plaque regression by lowering the intraplaque lipid deposition in Apoe^-/-^ mice fed with a high-fat diet (HFD) [128].

PLGA-NPs have also been successfully employed to influence the clinical and pathological course of infarcted hearts. Inflammation occurring in MI was investigated by Bei et al. [129], who demonstrated the cardioprotective effect of the flavonoid wogonin (5,7-dihydroxy-8-methoxy flavone) loaded in PLGA (WOG-NPs) in a rat model of MI induced by isoproterenol (ISO). Administration of WOG-NPs for three weeks before ISO poisoning induced a marked dose-dependent decrease in the infarct size in comparison with the ISO-induced MI group. Inflammatory parameters, such as IL-1β, IL-6, and TNF-α, were down-regulated. Moreover, the lipid peroxidation marker MDA was up-regulated in the ISO-induced MI model, but its content was reduced in the WOG-NPs treated group. The antioxidants enzymes CAT and SOD were significantly down-regulated in the ISO alone rats, while their values increased in a dose-dependent manner when the rats were treated with WOG-NPs. Similarly, PLGA-NPs have also been exploited to assess the anti-inflammatory effects of (E)-2-benzylidene-3-(cyclohexylamino)-2,3-dihydro-1Hinden-1-one (also known as BCI) in experimental MI induced by left anterior descending (LAD) coronary artery ligation. BCI is a chemical compound with anti-inflammatory properties but characterized by poor solubility and low bioavailability, thus limiting its pharmacological use. However, the BCI-loaded PLGA-NP administration effectively inhibited cardiac inflammation by decreasing macrophage activation and cytokine release in experimentally infarcted rats. Also, BCI-loaded PLGA-NPs reduced cardiomyocyte death and abnormal tissue remodeling, improving heart function in the same animals [130]. Furthermore, PLGA-NPs have been tested to improve tissue repair after MI. With this aim, Hu S et al. demonstrated that the intramyocardial injection of PLGA-NPs carrying mesenchymal stromal cell-secreted factors (MSCFs) improved cardiac function by protecting cardiomyocytes from ROS injury and apoptosis, reducing fibrosis during the cardiac remodeling process, and restoring myocardium volume [131]. Currently, autologous adipose-derived stem cells (ADSCs)-based therapies are under investigation in clinical trials as a therapeutic strategy for ischemic heart diseases due to their cardiac regenerative capacities [132]. Yokoyama R et al. investigated the possibility of treating infarcted mice with autologous adipose-derived stem cells (ADSCs) combined with simvastatin-conjugated PLGA-NPs. The authors revealed that the combined approach was more effective than the single treatments in stimulating cardiac regeneration and reducing the infarct site area [132].

## 5. Transplant Rejection

PLGA-NPs have also been used to control immune rejection in allograft transplantation, in which several severe side effects are associated with the use of immunosuppressive drugs. To avoid these side effects, new delivery systems targeting selected tissues/organs can limit systemic toxicity and improve the maintenance of the therapeutic concentration of the drug [133,134]. Currently, tacrolimus (TACR; FK506) is the most widely used immunosuppressive drug with anti-inflammatory properties to treat solid organ transplant rejection. TACR is a macrolide molecule that binds to cytoplasmic binding protein 12-FKBPl2; the formation of the complex suppresses the production of various cytokines and activation of T cells. TACR is routinely administered orally or intravenously, but its use is limited by its narrow therapeutic window, poor solubility, and low bioavailability [135]. Therefore, transplant patients require high dosages to maintain an effective concentration, increasing the risk of adverse drug reactions, including nephrotoxicity, opportunistic infections, and cancers. Moreover, the long-term use of TACR carries high economic costs [136]. The route and method of administration can significantly influence the bioavailability of TACR, and several pharmaceutical technologies, including nanomedicine, have been explored to optimize drug delivery, prolong survival time, and reduce the side effects [137].

Deng et al. designed a system that allows subcutaneous administration of TACR bypassing the drug’s bioavailability and toxicity issues using NPs of PLGA-TACR to improve TACR water solubility and to ensure its sustained release [138]. In rat cardiac rejection, subcutaneous injection of PLGA-TACR NPs reduced the blood concentration of TACR, which was stable for 10 days, decreased adverse effects, and significantly prolonged the rat survival time [138].

TACR is also prescribed in ocular therapy to inhibit corneal immunological rejection after keratoplasty. However, its conventional formulation limits its applications due to its hydrophobicity and low corneal penetrability, as well as its elimination by tear irrigation and nasolacrimal duct drainage. Wu et al. developed NPs based on an amphiphilic copolymer consisting of methoxy PEG (mPEG) and PLGA that exhibits high bioavailability and stability loaded with TACR. In rats with allogenic penetrating keratoplasty, administration of eye drops of these TACR-NPs increased TACR concentrations in the aqueous humor and cornea and inhibited IL-2, IL-17, and VEGF expression better than conventional 0.1% TACR eye drops [139]. Liu et al. developed a similar platform, TACR loaded in mPEG-b-PLGA micelles, that significantly inhibited rejection of corneal allograft in rats by lowering expression of nuclear factor of activated T cells (NFAT) and infiltration of CD4+ and CD8+ T cells [140].

To produce alloantigen-presenting NPs able to induce tolerance, Shahzada et al. created PLGA NPs coated with the H-2Kb-Ig construct to target H-2Kb alloantigen-specific CD8+ T cells, together with anti-Fas mAb, PD-L1-Fc, and TGFβ delivering negative signals to T cells. Furthermore, CD47-Fc served as a “don’t eat me” signal to inhibit NPs uptake by phagocytes. These NPs were highly effective in inhibiting rejection in a mouse model of allogeneic skin transplantation [141].

Li et al. developed PLGA microparticles loaded with TGF-β1 to prevent transplant rejection in allogeneic islet transplants [142]. The basic idea was to exploit the local release of TGF-β1 to create an immunotolerant environment favoring the engraftment of the donor’s pancreatic islets in the recipient [143,144,145,146]. However, soluble TGF-β1 has a short half-life and can induce a variety of side effects outside the target sites, including fibrosis [146]. The TGF-β1 PLGA microparticles enable localized and controlled release of TGF-β1 within the transplant site. Particles were fabricated at the microscale to avoid phagocytosis or convective clearance by phagocytes, thus increasing their retention at the graft site. In a diabetic murine model, co-injection of the TGF-β1 PLGA microparticles with allogeneic islets induced a significant increase of infiltrating T regulatory cells in the peri-transplantation site compared to the absence of the microparticles, but it did not significantly inhibit rejection [142] (Figure 2).

Another promising application of PLGA technology in the prevention of transplant rejection is the microencapsulation of transplanted pancreatic islets. Microencapsulation is a method of immunoisolation capable of significantly increasing the survival of beta cells in the recipient patient. It has been proposed to use biocompatible polymers to create an ultrathin layer on the surface of the pancreatic islets, able to inhibit recognition and infiltration by immune cells and to promote the transport of nutrients and essential proteins. Pham et al. encapsulated the pancreatic islets by coating them with PLGA-PEG NPs surface-functionalized with 3,4-dihydroxyphenethylamine (DOPA) and loaded with TACR. DOPA enables covalent binding to pancreatic islet collagen, forming a protective shield that prevents islet recognition and immune infiltration. Furthermore, a sustained release of TACR from the NPs into the microenvironment surrounding the islets actively inhibits the activation of immune cells. In a diabetic mouse model, this approach induced a significant improvement in the survival time of transplanted islets [147] (Figure 2).

**Figure 2 pharmaceutics-15-01772-f002:**
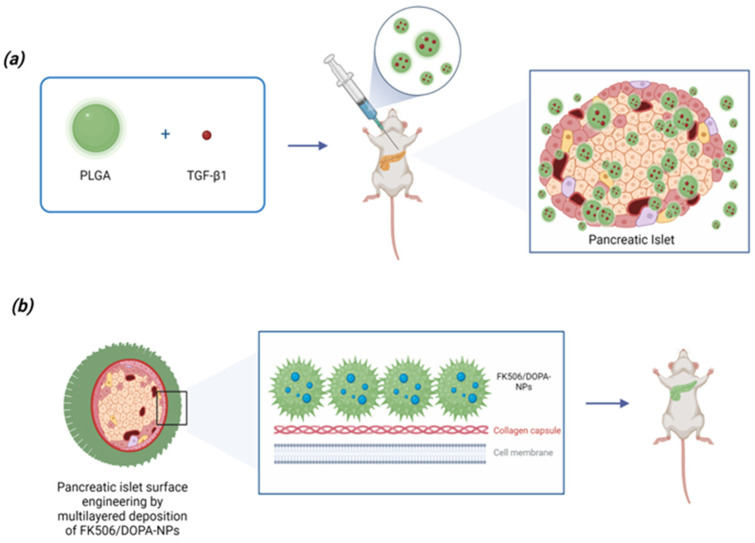
Application of PLGA nanoparticles technology for transplant rejection prevention. (**a**) Development of PLGA TGF-β1 microparticle system to prevent transplant rejection in allogeneic islet transplant. Incorporation of TGF-β1 in a PLGA microparticle system can support the localization and controlled release of TGF-β1 in the transplant site [143]. (**b**) Microencapsulation of transplanted pancreatic islets: the pancreatic islets are encapsulated by a coating of PLGA poly(ethylene-glycolide) (PLGA-PEG) nanoparticles, surface-functionalized with 3,4-dihydroxyphenethylamine (DOPA) and carrying the immunosuppressant TACR in the core. DOPA can bind pancreatic islet collagen forming a protective coating that can prevent islet rejection [148]. This image was created with BioRender.com (accessed on 20 April 2023).

## 6. Neurologic Diseases

Neurodegeneration is associated with a variety of disorders affecting the central nervous system (CNS), such as Alzheimer’s disease (AD), Parkinson’s disease (PD), multiple sclerosis (MS), epilepsy, and Huntington’s disease [148,149]. Besides MS, which is mainly driven by autoimmune mechanisms, the etiologic and pathogenic mechanisms of these diseases are still mostly elusive, but a contribution of neuroinflammation has been suggested in many cases. Treatment for these disorders is challenging due to blood–brain barrier (BBB) permeability restrictions; however, drug-loaded nanosystems have provided promising strategies to overcome this limitation, with particular attention given to PLGA-NPs [150,151,152].

Many theories have emerged to explain AD pathogenesis, and the contribution of inflammation is one of them. Glial cell activation, particularly in advanced disease stages, contributes to neuronal toxicity and cell death [153,154]. Long-term treatment with non-steroidal anti-inflammatory drugs (NSAIDs) such as ibuprofen is associated with adverse effects and poor biocompatibility in human clinical trials [155]. To overcome this problem, Sanchez-Lopez et al. developed PEGylated PLGA NPs loaded with dexibuprofen (DXIBU), the active enantiomer of ibuprofen, applied in a mouse model of AD. DXIBU-loaded PEG-PLGA NPs showed greater efficacy in preventing glial cell activation and Aβ accumulation than its free form, resulting in inhibiting neuroinflammatory responses caused by microglia [156]. In a recent study, the antioxidant ascorbic acid and epigallocatechin gallate (EGCG) were co-encapsulated in PEG-PLGA NPs in a transgenic model of AD, and the treatment showed that the efficacy was 5-fold higher when compared to free EGCG, resulting in decreased neuroinflammation and Aβ plaques formation [157]. Vilella et al. developed zinc-loaded PLGA NPs conjugated with a BBB penetration peptide for efficient targeting. This study showed that zinc-loaded PLGA NPs reduce the level of neuroinflammatory cytokines and Aβ plaque burden in an AD mice model [158]. Delivery of vitamin D-binding protein (DBP) encapsulated in PLGA was formulated by Seong et al. to effectively inhibit Aβ aggregation and neuroinflammation. These results show that DBP-PLGA NPs significantly inhibit Aβ aggregation and neuroinflammation in both in vitro and in vivo models [159]. Other authors suggested that the NPs loaded with the polyphenol CURC may be a promising drug delivery strategy to protect neurons against oxidative damage in AD [160]. These findings have been confirmed In vivo since PLGA-NPs loaded with CURC and an inhibitor of Aβ generation (PQVGHL peptide) decreased the level of Aβ burden, inflammation (TNF-α and IL-6), and oxidative stress markers (ROS) and enhanced the activity of the antioxidant SOD. These biochemical events came along with the increase of synapses in AD mouse brains and the attenuation of cognitive impairments [161]. In an AD rat model, a CURC-PLGA-NPs formulation was able to release CURC at constant rates and effectively induced neuronal differentiation by enhancing pro-neurogenic genes [162]. Another oxidative-stress-responsive PLGA-based NP was synthesized by Lei et al. [163] to deliver RAP, a specific inhibitor of mammalian RAP target protein (mTOR) with a recognized ability to promote degradation of amyloid *β* (A*β*), in both in vitro and in vivo models of AD. The ROS responsivity was achieved by modifying PLGA with Polyol-ox. Since AD tissues show high levels of oxidative stress, this ROS-responsive release system allowed to target the AD lesions more precisely, avoiding damage to the surrounding normal tissues. 

Epilepsy has a relatively large number of treatments available, but many patients develop resistance to conventional antiepileptic drugs (AEDs) [164]. A promising example of novel epilepsy treatments was performed by Cano et al., showing that treatment with EGCG encapsulated in PEGylated PLGA of a mouse model of temporal lobe epilepsy substantially decreased neuronal death and neuroinflammation [165]. 

PD is characterized by the degeneration of the dopaminergic neurons resulting in dopamine loss [166]. Delcroix et al. used PLGA microspheres coated with laminin and poly-D-lysine and releasing neurotrophin 3 (NT3) as a 3D scaffold to improve the survival of multipotent mesenchymal stromal cells implanted in the striatum of hemiparkinsonian rats. This treatment induced strong neuroprotection/repair of the dopaminergic nigrostriatal pathway, promoting neuronal differentiation and survival [167]. In another study, microcapsules of PLGA loaded with vascular endothelial growth factor (VEGF) and glial cell line-derived neurotrophic factor (GDNF) demonstrated neuroregenerative effects in a PD mouse model [168]. Moreover, treatment of non-human PD primates with PLGA microparticles loaded with GDNF increased the CDNF half-life and improved the restoration of dopaminergic neuron function. Rasagiline mesylate (RM), an inhibitor of monoamine oxidase B (MAOB), is used as a first-line agent for early treatment of PD, but its effectiveness is weakened by water solubility hampering crossing of BBB, low oral bioavailability, and rapid elimination requiring frequent dosing. To overcome these limitations, Bali et al. [169] fabricated transdermal films of gellan gum in which PLGA RM-NPs were embedded. Transdermal delivery of PLGA RM-NPs showed an enhanced brain bioavailability compared to oral administration, and treatment of PD rats showed accumulation of dopamine and an increase of the antioxidant content, such as CAT and GSH [169].

Medina et al. developed PEGylated PLGA NPs loaded with adapalene and showed their effectiveness in the SOD1G93A transgenic mouse model of amyotrophic lateral sclerosis with the improvement of neuroprotection and decrease of inflammation [170].

Neurological dysfunctions following hemorrhage were investigated by Zang et al. [171] by assessing the protective role of CURC-PLGA NPs in an animal model of experimental subarachnoid hemorrhage (SAH). Besides the protective effects on the BBB and the inhibition of the inflammatory response, these scientists demonstrated that CURC-NPs also suppress SAH-mediated oxidative stress. In particular, they found that CURC-NPs prevented the increase of ROS, MDA, 3-NT, and 8-OHDG and the decrease of SOD, GSH-Px, and CAT activities occurring after SAH [171]. 

PLGA NPs have also been tested for trans-nasal delivery of drugs to increase specific delivery to the central nervous system [172]. In a rat AD model, PLGA NP loaded with memantine acting on NMDA receptors conferred protection against spatial memory deficits [173]. In PD models, lactoferrin-PLGA-PEG NPs loaded with the dopamine agonist or PLGA NPs loaded with L-DOPA decreased encapsulation in PLGA NPs, and nasal administration also improved the clinical picture with prolonged activity [174,175]. In a model of cerebral ischemia, PLGA NPs loaded with the natural anti-inflammatory agent glycosyloxyflavone BA attenuated neuroinflammation [176]. In rat seizure models, chitosan-conjugated PLGA NPs loaded with the polyphenolic flavonoid catechin hydrate displayed anti-seizure effects [177]. The plant flavonoid baicalin (BA) was selected as the neuroprotective agent in a rat model of focal cerebral ischemia (MCAO). BA was loaded in PEG–PLGA NPs modified with the rabies virus glycoprotein (RVG29), a peptide that serves as a brain-targeting ligand. Using the nose-to-brain-based delivery of this nanomedicine, the neurological dysfunction of rats with ischemic brain injury was significantly alleviated with concomitant suppression of the inflammatory (IL-1b, IL-6, TNF-a) and oxidative stress, with a decrease of MDA level and an increase of the antioxidant defenses, such as GSH and SOD [176].

PLGA has also been used to set up devices to support peripheral nerve regeneration. The biomaterials should be biocompatible and biodegradable, allow adequate cell adhesion, display sufficient porosity and adequate mechanical properties, and, likely by offering aligned topography, may favor directional outgrowth of neurites [178]. In the rat sciatic nerve regeneration model, nerve guidance conduits (NGC) composed of aligned electrospun PLGA microfibers allowed better regeneration than random microfibers [179]. Other authors improved the adhesive properties of a similar system using mussel adhesive peptides and IKVAV, a peptide derived from laminin [180]. Wang W.C. showed accelerated restoration of motor function in injured sciatic nerve using nerve guidance conduits based on gelatin/silk hydrogel, including linearly aligned magnetic PLGA microcapsules encapsulating nerve growth factor [181]. Zhang J. obtained a similar positive effect using collagen-coated PLGA conduits loaded with insulin-like growth factor 1 modified by a collagen-binding domain [182], while Chung Y.F. obtained a similar effect using PLGA conduits releasing IL-12 [183]. Post-traumatic inflammation was investigated in an animal model of brachial plexus avulsion (BPA) [184], a severe injury involving the central and peripheral nerves characterized by oxidative damage and inflammatory reactions leading to widespread neuronal death and nerve regeneration difficulty [185]. QC was loaded in a temperature-sensitive PLGA-PEG-PLGA hydrogel and then injected into an animal model of BPA to alleviate oxidative stress and inflammation. Interestingly, along with inhibition of ROS and IL-6 production, QC PLGA-based hydrogel increased neuron survival rate and promoted nerve regeneration and motor function recovery in rats with early BPA [184]. 

A topical approach has also been used to control chronic neuropathic pain. Zhang et al. detected a substantial analgesic effect on pain hypersensitivity induced by spinal nerve ligation in mice by injection around the nerve root of a nanodevice composed of PLGA NPs encapsulating esketamine, a non-opioid analgesic causing severe side effects, embedded in hyaluronic acid hydrogel [186]. 

## 7. Osteoarthritis and Bone Regeneration

Therapy for osteoarthritis (OA) involves treatments that often have limitations, such as low solubility and poor bioavailability, degradation by gastrointestinal enzymes, and toxicity [187,188]. To overcome these limitations, systems have been developed to deliver drugs based on the use of PLGA NPs for intra-articular applications to penetrate the joint layers and reach the sites of inflammation [189]. The small molecular drug salicin (SA) loaded into PLGA NPs was administered intra-articularly in a rat model of OA induced by anterior cruciate ligament transection (ACLT) [190]. SA-PLGA NPs were shown to be highly effective at reducing OA progression by inhibiting inositol-requiring enzyme 1α (IRE1α)-mediated endoplasmic reticulum (ER) stress, compared to empty PLGA. Moreover, they inhibited cartilage matrix degeneration and chondrocyte apoptosis. PLGA NPs loaded with meloxicam (MLX), a drug for the treatment of OA causing gastrointestinal side effects, were tested in a model of OA induced in male Wistar rats. The results showed that intra-articular injection of MLX-PLGA NPs can reach the site of inflammation after the injection into the joint cavity, reducing the risk of intestinal side effects [191]. Furthermore, PLGA-based scaffolds loaded with icariin, a component extracted from *Epimedium*, were tested in rabbit OA models [192] and showed positive effects in the resolution of the disease. The results obtained by X-ray microtomography and histological analysis demonstrated that the PLGA-icariin scaffold is effective at maintaining the morphology of the articular cartilage and inhibiting the resorption of the subchondral bone trabeculae of knees compared to PLGA alone [192]. Another approach used PLGA microspheres loaded with sulforaphane (SFN), a member of the isothiocyanate family with anti-inflammatory properties [193], in a rat model of OA. The intra-articular treatment could reduce inflammation and OA progression [194]. Moreover, intra-articular delivery of PLGA microspheres loaded with rapamycin (RAP) mitigated cartilage damage and inflammation in OA mice [193,195] (Figure 3). PLGA NPs loaded with siRNA to p16INK4, an inhibitor of cyclin-dependent kinases (CDK), were used in a partial medial meniscectomy induced-mouse OA model [196]. The tested system reduced the levels of TNF-α, IL-1β, and IL-6 in fibroblast-like synoviocytes and MMP13 in chondrocytes, with a reduction of the clinical manifestations and cartilage damage. Strategies aimed at lowering oxidative stress have also been explored. For instance, a rat model of OA was treated with PLGA NPs loaded with siRNAs specific for p47phox [197], a cytosolic subunit of NADPH oxidase involved in the production of ROS. The treatment showed improvement of OA by promoting ROS decrease and preventing chondrocyte cell death. Other authors used rebamipide (REB), which is able to attenuate pain severity and cartilage degeneration by downregulating oxidative damage and catabolic activity in chondrocytes in OA rats. In vitro, REB PLGA NPs were able to reduce the levels of IL-1β, IL-6, TNF-α, MMP-3, MMP-13, as well as COX-2 levels. In vivo, gross, radiographic, and histological evaluations showed that intra-articular injection of REB PLGA NPs inhibits cartilage degeneration compared to rats treated with free REB [198,199].

Alternative approaches used PLGA-based systems to deliver endogenous components such as platelets and microvesicles. Li et al. developed a super-activated platelet lysate (sPL) loaded in a PLGA-based system for the treatment of OA [200]. In vitro results showed that exposure to sPL-PLGA microparticles increases chondrocyte proliferation and reduces cell necrosis in osteoarthritic chondrocytes. In vivo experiments in a rat OA model showed that sPL-PLGA microparticles increase cartilage integrity. Other authors used PLGA NPs decorated with microvesicles derived from CD90^+^ synovial mesenchymal stem cells (MSC) and loaded with the glucocorticoid triamcinolone acetonide (TA) [201]. In a rabbit OA model, treatment with these NPs improved the repair of damaged cartilage compared to treatments with TA or MCS microvesicles alone. In particular, prolonged treatment restarted the cell cycle and reduced apoptosis of chondrocytes via the FOXO pathway and favored M2 macrophage polarization and IL-10 secretion. PLGA-based systems have been exploited to promote sustained drug release, as is the case of PLGA loaded with Diacerein (DIA), a slow-acting drug for OA [202]. In vitro studies with rat synoviocytes showed that treatment with DIA-PLGA reduces the levels of pro-inflammatory cytokines. Moreover, intra-articular treatment of OA rats with PLGA-DIA reduced the levels of pro-inflammatory cytokines and increased the anti-inflammatory ones, and decreased cartilage degradation.

The search for valid artificial compounds for the production of bone substitutes is constantly evolving. However, the materials must have good mechanical resistance with a porous structure to allow the cells to colonize and grow and guarantee biocompatibility and biodegradability [203,204,205,206,207]. Notoriously, the materials preferred by researchers have been hydroxyapatite (HA) [208,209], biphasic calcium phosphate [210,211], tricalcium phosphate [212,213], carbon nanotubes [214,215], or bioglass [216]. All these inorganic materials have shown good biocompatibility but also extreme fragility, which does not guarantee that they have adequate bone strength [217]. Collagens, too, received particular attention together with fibrin glues due to their good plasticity qualities [218,219], but they have low mechanical strength and are subject to easy degradation by collagenases. Unlike all these, synthetic polymers can be controlled both in terms of molecular mass and degradation time while maintaining excellent workability and good mechanical qualities. The resulting artificial substitutes have high bone repair rates [220,221]. PLGA itself has been used for bone repair because of its excellent biocompatibility and biodegradability [16,222] and the possibility to control the rate of degradation by modulating the ratio between glycolic and lactic acid to match the rate of bone regeneration [223]. Moreover, its progressive degradation gradually increases the stress on the newly formed tissue, progressively increasing its regeneration and bone remodeling. In addition, PLGA grafts can be loaded with bioactive factors, drugs, or stem cells to promote or speed up bone regeneration [224,225]. PLGA has excellent processability [226,227], favoring its use in several conditions leading to primary or secondary bone defects such as trauma [228], osteoporosis [229,230], bone tumors [231,232], infections [233], osteonecrosis [234], and iatrogenic lesions [235].

A wide application may be bone loss in the oral cavity that may be caused by bacterial periodontitis dealing with the erosion of the alveolar bone, periodontal ligaments, and the surrounding soft tissues [236] (Figure 3). The bone loss is also due to a lack of masticatory function at the site of tooth loss, as bone is a mechanosensitive tissue and suffers from disuse atrophy [237]. These problems can be amplified by systemic pathological conditions, such as diabetes mellitus [238]. The flat bones of the jaws and skull can generate new bone by differentiating MSC into osteoblasts [239], but the process is twice as slow compared to that of long bones [240]. Since bone autografts and allografts may display poor healing potential, a solution may be the use of synthetic biomaterial grafts specifically developed to promote the regeneration of maxillary and mandibular bone tissues. Encouraging results have been obtained with porous gelatin-hydroxyapatite cryogel infused with PLGA microspheres loaded with osteogenic factors facilitating alveolar ridge augmentation in vivo [241]. Zoledronate (ZOL), a bisphosphonate compound used to counteract osteoporosis [242], seems a promising agent. Indeed, Yao et al. investigated the protective role of ZOL-PLGA (PLGA-ZOL) microcapsules in a model of rat periodontitis [243], showing that PLGA-ZOL treatment significantly attenuates alveolar bone loss and suppresses oxidative stress by enhancing SOD and CAT activity in the gingival tissue. Moreover, PLGA-ZOLs inhibit pro-inflammatory cytokine production (TNF-α, IL-1β) and increase anti-inflammatory cytokine secretion (IL-10).

There is also great interest in the regeneration of the whole tooth. Here, too, PLGA may play an important role in the structure of the scaffolds and the release of bioactive molecules. Technologies to regenerate essential tissue components of the tooth rely on the use of stem cells, growth factors, and scaffolds [244]. Dental pulp stem cells have properties of multipotent differentiation, as well as self-renewal and clonogenicity, with a marked odontogenic propensity. They can differentiate in vitro into cells very similar to osteoblasts and create dentin/dental pulp structures in vivo [245]. In the processes of survival, proliferation, differentiation, polarization, and migration of these cells, a predominant role is played by the extracellular matrix [246]. All the new structures composed of biomaterials try to imitate the extracellular matrix [247]. The tooth root in mammals is a type of environment composed of several soft and hard tissues with distinct architectural and biochemical characteristics, including soft tissues such as the dental pulp and the periodontal ligament and hard tissues such as cementum and dentin. The periodontal area, where the cementum, the ligament, and the alveolar bone are present, is the most important for the possible regeneration of the tooth root and its support [248]. One of the technologies tested has been the cell sheet [249], where a sheet of cells is in contact with the extracellular matrix favoring the integrity of the whole structure before transplantation [250,251]. The sheet is generally an electrospun biomimetic with a nanofibrillated structure very similar to a normal extracellular matrix and acts as a scaffold [252]. An interesting model involved an aligned electrospun PLGA/gelatin sheet in simulating the extracellular matrix of the periodontal ligaments with parallelly aligned fibrils [253]. In vivo results suggested the validity of electrospun PLGA/gelatin sandwich constructs with treated dentin matrix and native dental pulp extracellular matrix, which allowed for regeneration of the root tissues of the tooth in miniature pig models [253]. 

An obstacle that can arise in the engineering of pulpal tissues using dental stem cells is the poor cell retention of the scaffolds. This aspect could be mitigated by injectable scaffolds, which are well-suited for irregular root canals, facilitating retention and clinical initiation. An interesting way of injecting the structures involves the use of cryogel gelatin methacrylate microspheres functionalized with simvastatin (SIM) to effectively seed stem cells from exfoliated human deciduous teeth directly into the root canals. SIM was encapsulated in PLGA NPs to allow the desired release from the scaffold, thereby enhancing stem cell biology and regenerating tissue and vasculature with characteristics close to the pulp [53,254]. SIM is a drug that is commonly used to treat hyperlipidemia, but it has demonstrated antioxidant and anti-inflammatory activities and the ability to promote tissue mineralization [255,256,257]. Its use made it possible to reduce the use of growth factors derived from platelets, fibroblasts, or vascular endothelium, which have high costs and a short half-life [258,259,260]. In these multilevel systems, SIM is delivered thanks to PLGA inside the methacrylate gel matrix and from there to the surrounding microenvironment [261].

## 8. Skin Wound Healing

Wound healing is a clinical problem in pathological conditions (i.e., diabetes) in which tissue repair is impaired [262] and may be improved by local treatment with biomaterials releasing factors capable of modulating inflammation and regeneration, which overcome the limitations of topical skin treatments represented by the rapid degradation of the bioactive compounds and need for frequent applications [263].

PLGA NPs encapsulating CURC [264] were employed in a mouse wound healing model and were shown to accelerate the repair rate compared to empty PLGA-treated mice or untreated mice. Histological and molecular analyses highlighted a greater re-epithelialization, formation of granulation tissue, and anti-inflammatory activity exerted by PLGA-CURC NPs [264]. With the same objectives, PLGA NPs containing heparin embedded in a hydrogel were tested on the wounds of diabetic and non-diabetic rats [265]. In non-diabetic rats, PLGA-heparin-containing hydrogel aided wound healing compared to free heparin-treated mice or untreated mice, while in diabetic mice, both free and encapsulated heparin promoted wound healing [265]. A hydrogel composed of PLGA PEG PLGA triblock copolymers was set up to deliver niobium carbide (Nb2 C) (Nb2C@Ge) with the intent to treat diabetic wounds that are challenging to treat due to the excessive levels of oxidative stress, vulnerability to bacterial infection, and persistent inflammation [266]. In an in vivo diabetic wound model, treatment with Nb2C@Ge showed antimicrobial activity and anti-inflammatory effects along with the suppression of oxidative stress. Similar results were obtained with CURC-loaded PLGA nanofiber membranes (PC NFMs), followed by high-density surface grafting of heparin [267] or with PLGA nanofibers electrospun with LPS/IFN-γ activated macrophage cell membrane [268]. In both cases, anti-inflammatory and antioxidant effects were concomitant with the acceleration of wound closure in diabetic rats. Tylotoin (TYL) is a peptide obtained by salamander and can favor wound healing by promoting migration and proliferation of endothelial cells, keratinocytes, and fibroblasts, recruitment of macrophages, and release of repairing cytokines, but its use is hampered by fast degradation and low transdermal delivery. Topical treatment of mouse skin wounds once every two weeks with chitosan PLGA NPs loaded with TYL was able to improve wound healing more effectively than daily treatment with TYL alone [269]. In ICR SCID mice, fibroblast growth factor 10 (FGF10)-encapsulated PLGA microspheres applied to wounds accelerated the wound closure rate and improved granulation tissue formation, collagen synthesis, cell proliferation, and vascular density, and inhibited expression of endoplasmic reticulum (ER) stress markers [270]. Electrospun nanofibers containing PLGA NPs encapsulating the dipeptidyl peptidase-4 (DPP4) inhibitor vildagliptin (VILD) were developed to promote wound healing closure by inhibiting this protease involved in the inactivation of several growth factors [271]. Using Sprague–Dawley streptozotocin-induced diabetic rats, the application of the membranes containing PLGA-VILD was effective in accelerating wound closure and promoting neo-angiogenesis [271]. A single dose of electrospun nanofibers containing PLGA NPs encapsulated with phenytoin (PH), an antiepileptic drug with known wound healing activity, was able to accelerate the skin repair process with optimal re-epithelialization profile without significant signs of inflammation or necrotic areas [272]. Neurotensin (NT) is a neuropeptide capable of modulating cell migration and chemokine production and has been found to be effective for diabetic wound healing [273]. Nanofiber membranes of PLGA and cellulose nanocrystals loaded with NT have been tested on the wounds of diabetic mice. They have been shown to release NT for two weeks, promoting rapid wound healing compared to control mice, accompanied by a decrease in proinflammatory cytokines [274].

Another approach has been to use PLGA devices to deliver cell-based therapies. The use of MSC is effective at repairing diabetic wounds. However, poor cell survival after transplantation into the wound is a frequent hurdle [275]. To overcome this problem, IL-8-containing PLGA NPs loaded onto an acellular dermal matrix were used as a delivery system for exogenous MSC. In diabetic mice, this treatment increased wound healing and favored capillary formation with increased production of proangiogenic factors and collagen deposition. Adipose-derived stem cells (ADSCs) combined with PLGA microspheres loaded with velvet antler polypeptide (VAP), known to favor cell migration and angiogenesis, was effective in a chronic skin ulcer model induced in immunosuppressed mice, in which this therapy increased ADSCs activity, promoting cell migration, angiogenesis, and collagen deposition, and in enhancing levels of VEGF, HIF-1a, p-Akt and p-PI3K [276]. Moreover, PLGA/gelatin fibrous membrane scaffolds containing hyaluronic acid (HA) [277] were used to deliver ADSCs in rat wound healing. The treatment was able to favor wound closure by accelerating healing times and increasing the epithelialization rate compared to the same system without ADSCs. The addition of ADCSs promoted the recruitment of reparative macrophages and angiogenesis without type I collagen deposition [277].

The use of PLGA particles has also been suggested for the treatment of atopic dermatitis, a complex multifactorial disease involving barrier dysfunction, impaired cell-mediated immune responses, IgE-mediated hypersensitivity, and environmental factors [278]. Dermatitis induced in mice by topical application of dinitrofluorobenzene was effectively treated with PLGA-mPEG microparticles loaded with the antihistamine mizolastine with inhibition of dermatitis and reduction of the inflammatory cell infiltrate and plasma IgE levels [279]. PLGA NPs loaded with dictamnine, an active pharmaceutical ingredient in *Dictamnus dasycarpus*, were effective in the treatment of oxazolone-induced atopic dermatitis by decreasing inflammation and expression of inflammatory cytokines [280]. Chitosan-coated PLGA particles loaded with ceramide restored the skin stratum corneum in rat experimental atopic dermatitis [281]. Moreover, PEG-PLGA microparticles loaded with IL-2, RAP, and TGF-β1 displayed topical immunomodulatory activity in mouse allergic contact dermatitis [282] since the treatment was able to locally expand T regulatory cells and decrease pro-inflammatory effector T cells.

Acne is a common skin disease caused by infection within the hair follicle. It is usually treated with antibiotics, resulting in imbalanced skin microbiota and microbial resistance. A key problem is related to the difficult penetration, retention, and release over time of the drugs applied to the skin [282]. Recently, PLGA NPs loaded with thymol (TH), a natural product with antibacterial and antioxidant properties, were shown to be effective for the treatment of cutaneous acne infections without the need to administer antibiotics [283]. The use of TH-PLGA NPs allowed the prolonged release of TH with high penetration and retention inside the hair follicle and enhanced the antimicrobial activity, especially against *Cutibacterium acnes* [284,285].

## 9. Eye Diseases

The static (corneal layers, retina, sclera, blood-aqueous and blood-retinal barriers) and dynamic (tear dilution, conjunctival and choroidal blood flow, and lymphatic clearance) barriers are the main causes of the challenges in the administration of ocular medications [286,287,288] because of the need to continuously maintain an effective medication concentration during treatment. Moreover, drug delivery through eye drops, injections, or repeated administration often has adverse effects such as endophthalmitis, cataracts, and retinal detachment [289,290]. In the treatment of eye diseases, PLGA microspheres offer good biocompatibility and safety and can be employed as a continuous ocular drug delivery method (Figure 4).

Barcia et al. examined the effectiveness of PLGA microspheres loaded with dexamethasone (DEX) to minimize inflammation induced by intravitreal injection of lipopolysaccharide (LPS) into rabbit eyes [291]. Treatment with DEX-loaded microspheres resulted in a rapid decrease in eye inflammation and resistance to a second challenge with LPS after 30 days [291]. Another work used the same animal model to assess the anti-inflammatory activity of chitosan-PLGA NPs loaded with triamcinolone acetonide (TA), a synthetic corticosteroid widely used in ocular inflammatory diseases since free drug suspension displays several adverse effects. When compared to free TA suspension, TA-loaded PLGA NPs significantly reduced the inflammatory score after a single dosage in a rabbit model; significant concentrations of TA peaked after 6 h and persisted at high levels for up to 24 h [292]. Ratay et al. used intralacrimal injection of PLGA microspheres loaded with the histone deacetylase inhibitor suberoylanilide hydroxamic acid (SAHA) to treat dry eye disease (DED), also known as keratoconjunctivitis sicca involving T cell-driven inflammation, to increase the expression of FoxP3, a transcription factor involved in the development of T regulatory cells. In mice, clinical signs of Concanavalin A-induced DED were reduced by this treatment, which also increased the expression of FoxP3 and decreased the expression of pro-inflammatory cytokines [293]. Gosh et al. demonstrated that the antioxidant flavonoid xanthohumol loaded in PLGA NPs inhibits oxidative stress in vitro and significantly reduces ocular surface damage and oxidative stress-associated DNA damage in corneal epithelial cells in the mouse desiccating stress/scopolamine model of dry eye disease in vivo [294]. In alkali burn-induced Sprague–Dawley rats, PLGA NPs loaded with the anti-fibrotic agent pirfenidone accelerated ocular wound healing and avoided fibrosis with positive effects on corneal re-epithelialization and scarring; treatment decreased the level of collagen I and the development of myofibroblasts [295]. In the rabbit corneal alkali burn model, treatment with PLGA NPs containing *Lingzhi* extracts (LZH-NPs), a Chinese herbal drug with anti-oxidative and anti-inflammatory properties, down-regulated inflammation and ROS content, decreased the area of corneal opacity, and accelerated the corneal wound healing [296].

Gambogic acid (GA), a natural xanthonoid capable of non-competitive binding to the transferrin receptor, has been used for eye-targeting of coumarin-loaded PLGA NPs given by oral administration since transferrin receptors favors transport across the intestinal and blood-eye barriers. In a canine in vivo model of ocular inflammation mimicking the uveitis following cataract surgery, oral administration of double-headed GA-coupled PLGA NPs loaded with coumarin was effective at reducing the intraocular inflammatory response assessed using an ocular scoring system [297].

Glaucoma represents a worldwide neurodegenerative disease that causes irreversible blindness. The administration of glaucoma drugs remains an unmet clinical need [298]. A system based on PLGA microspheres incorporating three neuroprotective agents (DEX, melatonin, and coenzyme Q10) in a single formulation was recently developed to create a novel intraocular delivery system for glaucoma treatment [299]. A rodent model of chronic ocular hypertension was used to compare single intravitreal injections of microspheres co-loaded with the three drugs versus those loaded with a single drug or the empty ones. After 21 days, a neuroprotective effect was detected in the group treated with the devices coloaded with the combination of the three drugs [299]. Another approach used PLGA NPs loaded with dorzolamide (DZ), a carbonic anhydrase inhibitor used in glaucoma therapy, and showed that a single instillation into the eye substantially reduces ocular pressure [300]. To favor ocular retention of DZ, mucoadhesive microparticles were used in the eyes of rabbits, showing a significant and prolonged decrease in ocular pressure [301]. PLGA microspheres were also loaded with growth factors. For instance, PLGA microspheres containing glial cell line-derived neurotrophic factor (GDNF) were developed to promote retinal ganglion cell survival, expressing both GDNF and its receptor, in a rat model of glaucoma. Intravitreal injection of GDNF-PLGA microspheres did not decrease ocular pressure, but it increased retinal ganglion cell and axonal survival and decreased retinal degeneration [302]. Since several reports showed that erythropoietin (EPO) reduces cell death, axon degeneration, and oxidative stress in several models of glaucoma [303], Naguib et al. [304] developed PLGA microparticles loaded with EPO-R76E, a mutant form of EPO preserving retinal neurons and optic nerve axons, without inducing a significant increase of the haematocrit. The intravitreal injection of PLGA.EPO-E76E in glaucoma mice displayed neuroprotective activity and prevented increases of retinal ^•^O_2_—levels and caused activation of the NRF2/ARE antioxidant pathway. Positive effects were also obtained in a mouse model using PEG-PLGA NPs loaded with memantine, approved for AD [305]. Ocular applications for three weeks were effective at reducing the loss of retinal ganglion cells [305]. Since a disadvantage of eye drops is that only small amounts reach the posterior segment, Sharma et al. developed collagen membranes embedded with PLGA-NPs, which keep contact with the cornea, ensuring continuous delivery of drugs to the target site [306].

## 10. Conclusions

Inflammation is a common pathogenetic mechanism underlying a large spectrum of pathological conditions. Its modulation may strongly impact the disease evolution toward advanced stages either by inhibiting the defensive/aggressive phase of inflammation, thus decreasing tissue damage, or by supporting the reparative phase, thus favoring tissue regeneration. Although several drugs are already available to inhibit or modulate inflammation, their use is often limited owing to the difficulty of maintaining effective therapeutic concentrations and the appearance of severe side effects that hamper their clinical applications for long periods. Moreover, the therapeutic use of natural bioactive molecules to substitute conventional drugs is made difficult by their physicochemical properties that negatively influence their bioavailability and, thus, their effectiveness in targeting inflammation. Overall, this review article examines PLGA-NPs as versatile drug carriers that overcome many of the limitations concerning drug administration, making classical and novel anti-inflammatory molecules efficacious against multiple inflammatory-related conditions. These positive effects on several typical inflammatory diseases, such as allergic and autoimmune diseases (Table 1), where they mainly act by inhibiting the defensive/aggressive phase of inflammation, have been widely reported. However, a greater number of studies show that PLGA-NPs can also be effective in other diseases caused by an imbalance between the defensive/aggressive phase and the reparative phase of inflammation, with positive effects not only in inhibiting the former but also in favoring tissue regeneration (Table 2). PLGA-NPs allow the expansion of the therapeutic armamentarium and enable site-specific drug delivery that holds promise in reducing off-target effects, thus also allowing long-term treatment in chronic inflammatory disorders.

## Figures and Tables

**Figure 1 pharmaceutics-15-01772-f001:**
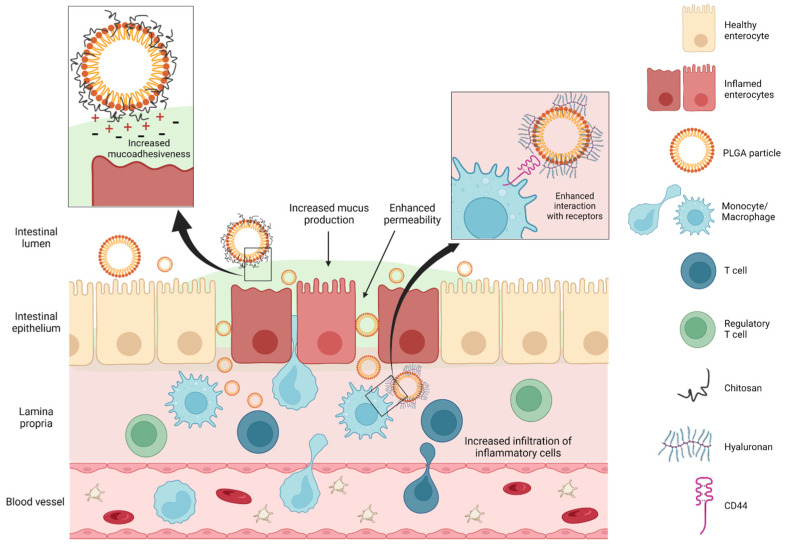
Mechanism of action of PLGA particles in IBD. The three key points in the PLGA particle mechanism of action in the inflamed intestine are enhanced permeability of the intestinal epithelium, increased mucus production, and greater infiltration of inflammatory cells in the *lamina propria*. The particle size influences its final destination and behavior, with large particles limiting their action to the apical surface of enterocytes and middle-sized and small particles targeting the inflammatory cells of the basolateral *lamina propria*. The two boxes display a magnification of functionalized PLGA particles. On the left, PLGA is complexed with the cationic polymer chitosan to enhance the particle mucoadhesiveness by improving the electrostatic interactions with mucus. On the right, a hyaluronan-functionalized PLGA particle interacts with a macrophage by binding to CD44, which recognizes hyaluronan. This image was created with BioRender.com (accessed on 20 April 2023).

**Figure 3 pharmaceutics-15-01772-f003:**
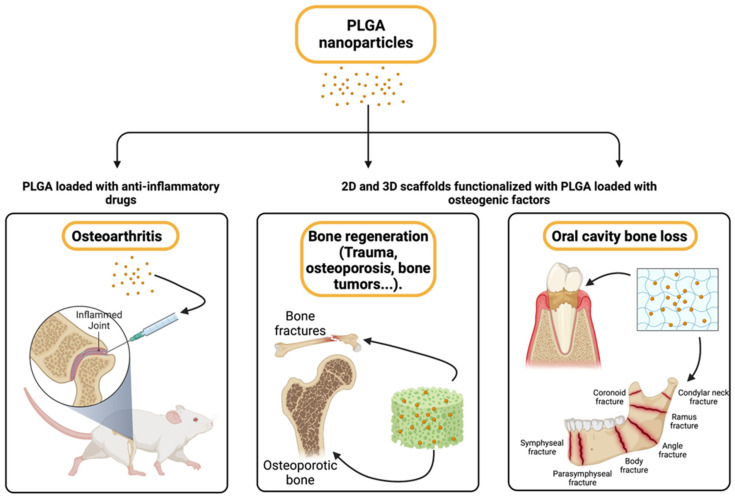
The image shows the use of PLGA in OA and bone regeneration field. PLGA nanoparticles loaded with anti-inflammatory drugs have been used to treat inflamed joints in OA animal models through intra-articular injections. In the bone regeneration context, different 2D and 3D scaffolds may be functionalized with osteogenic factors loaded in PLGA particles. This image was created with BioRender.com (accessed on 20 April 2023).

**Figure 4 pharmaceutics-15-01772-f004:**
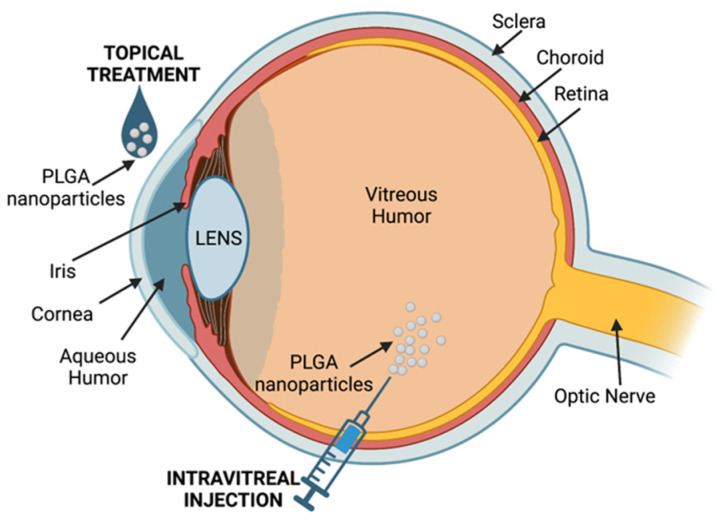
Schematic representation of the ocular structure with various ocular barriers and PLGA route of administrations. Both intravitreal and topical treatments have to pass different barriers, in particular, the sclera, the choroid, and the vitreous humor. In addition to these barriers, the topical treatment must also pass the tear film, the corneal epithelium, and the aqueous humor. This image was created with BioRender.com (accessed on 20 April 2023).

**Table 1 pharmaceutics-15-01772-t001:** Application of PLGA NPs in immunotherapy [31].

Disease		Cargo Molecules
Allergic diseases	Allergic Asthma	Antisense oligonucleotides targeting Dnmt3, chrysin, OVA, A20, CpG, Der p2
	Allergic conjunctivitis	Recombinant Amb a 1
	Cow’s milk allergy	β-lactoglobulin derived peptides, β-lactoglobulin
	Honey bee venom allergy	PLA2, protamine stabilized CpG
Autoimmunity	Multiple sclerosis	LIF, MOG_35–55,_ IL-10, TGF-β, GM-CSF, Glucosamine-MOG_35–55_, PLP_139–151_
	Guillain-Barrè Syndrome	Cargo-free
	Type I diabetes	B_9–23_, TGF-β, GM-CSF, Vitamin D3, Denatured insulin, CM-GSF, CpG
	Rheumatoid arthritis	CII, Altered CII AP_268–270_, Tacrolimus, Cargo-free
	Systemic lupus erythermatosus	miRNA-125a, TGF-β, IL-2
Infectious diseases	Viral hepatitis	HBV nucleocapsid core antigen, MPLA, HBsAg, HCV E2 envelope glycoprotein
	Influenza	HA (H5N1) imiquimod, Inactivated H4N6, CpG, Capsomere presenting M2e, M1_58–66_, PA_46–54_, CpG, polyIC, OVA
	HIV	HIV-1 gp120, PEI/plasmid-DNA complex (Gag, Pol, Env)
	COVID-19	Recombinant SARS-CoV-2 S1-E, Trivalent SARS-CoV-2 subunits
Cancer immunotherapy	Immune checkpoint inhibition	Cargo-free
	PTT + immune checkpoint inhibition	Anti-PD1 mAb, Imiquimod, ICG
	Modulation of tumor microenvironment	Oxaliplatin, R848, Paclitaxel, CXCL-1
	Modulation of tumor microenvironment + immune checkpoint inhibition	Metformin, siRNA FGL1
	Cancer vaccine	OVA mRNA, gardiquimod, 4T1 cell lysate, NY-ESO-1 TAA, IMM60
	Cancer vaccine + immune checkpoint inhibition	Imiquimod, OVA, HLA-A*0201 restricted TAA, Riboxxim, HPV-E7, siRNA PD1, siRNA PD-L1

**Table 2 pharmaceutics-15-01772-t002:** Use of PLGA NPs to improve tissue regeneration in experimental models.

Disease	Drug	**Citation**
Intestinal Bowel Disease	Phellinus igniarius	[86]
Myocardial infarction	(E)-2-benzylidene-3-(cyclohexylamino)-2,3-dihydro-1Hinden-1-one (BCI)	[117]
	Mesenchymal stromal cell-secreted factors (MSCFs)	[118]
	Simvastatin (plus adipose-derived stem cells	[119]
Parkinson disease	Neurotrophin 3 (NT3)	[154]
	Vascular endothelial growth factor (VEGF) plus glial cell line-derived neurotrophic factor (GDNF)	[155]
Peripheral nerve damage	Mussel adhesive peptides and IKVAV peptides	[167]
	Nerve growth factor	[168]
	Insulin-like growth factor	[169]
	IL-12	[170]
	Quercetin	[171]
Osteoarthritis	Super-activated platelet lysate (sPL)	[187]
	Synovial mesenchymal stem cells microvesicles	[188]
Bone loss	Zoledronate	[229]
	Simvastatin	[241,242]
Wound healing	Curcumin	[253]
	Niobium carbide (Nb2 C)	[254]
	Curcumin plus heparin	[255]
	LPS/IFN-γ activated macrophage cell membrane	[256]
	Tylotoin	[257]
	Fibroblast growth factor 10 (FGF10)	[258]
	Vildagliptin	[259]
	Phenytoin	[260]
	Neurotensin (NT)	[261,262]
	IL-8 plus mesenchymal stem cells	[263]
	Velvet antler polypeptide plus adipose-derived stem cells	[264]
	Hyaluronic acid (HA) plus adipose-derived stem cells	[265]
Ocular wound healing	Pirfenidone	[283]
	Lingzhi extracts	[284]

## Data Availability

No new data were created or analyzed in this study. Data sharing is not applicable to this article.

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
