# Peer review of "Use of Poly Lactic-co-glycolic Acid Nano and Micro Particles in the Delivery of Drugs Modulating Different Phases of Inflammation"

_pharmaceutics, 2023, doi:10.3390/pharmaceutics15061772_

Round 1
Reviewer 1 Report
This paper overviewed the PLGA nano and micro particles in delivery of drugs modulating different phases of inflammation. It is interesting. However, prior to consideration of publication, some revisions need to be addressed.
1. The subsection 2 the description about Oxidative Stress is a bit to much, and it is recommended to reduce them.
2. The authors devote a great deal of words to describing disease and the use of PLGA in immunity, but rarely relate these to the different stages of inflammation.
3. Line146” to contrast oxidative stress levels”, the meaning is not clear.
4. There are some problems in the use of present and past tenses in the text.
5. This paper presents a lot of data, and more discussion and analysis of data correlation and significance should be added.
Minor editing of English language required
Author Response
REVIEWER 1
This paper overviewed the PLGA nano and micro particles in delivery of drugs modulating different phases of inflammation. It is interesting. However, prior to consideration of publication, some revisions need to be addressed.
- The subsection 2 the description about Oxidative Stress is a bit to much, and it is recommended to reduce them.
The section has been reduced by about 20%
- The authors devote a great deal of words to describing disease and the use of PLGA in immunity, but rarely relate these to the different stages of inflammation.
Use of PLGA nanoparticles to support the repairing phase of inflammation is cited in the text whenever is the case. To highlight this alternative form of immunomodulation, different from immunosuppression, we created the novel Table 2 summarizing the several uses of PLGA NPs to support tissue repair described in the review
- Line146” to contrast oxidative stress levels”, the meaning is not clear.
The sentence has been changed to ”…to contrast oxidative stress”
- There are some problems in the use of present and past tenses in the text.
The English form has been revised
- This paper presents a lot of data, and more discussion and analysis of data correlation and significance should be added.
The Introduction section has been extended to improve the description of the different types of PLGA NPs. The Conclusion section has been extended to improve the data wrapping up
Comments on the Quality of English Language
Minor editing of English language required
The English form has been revised
Reviewer 2 Report
As a drug carrier with high tissue compatibility and strong operability, PLGA-NPs has been widely used. This review summarizes the application of PLGA-NPs in some inflammatory diseases and illustrates the principles of its possible role. However, there are still some problems that need to be solved.
1. The article lists IBD, skin injury diseases, etc., but it does not cover some other common inflammatory diseases, such as pneumonia, asthma, etc. Can PLGA-NPs be applied to these diseases, or is it not reported? The authors had better summarize a table that can clearly show which diseases the PLGA-NPs delivery system has been applied.
2. Does the targeted delivery of PLGA-NPs mainly rely on particle size? Are there any modifications that can increase their organ-specific targeting? If so, please summarize the different modification methods and the principles in the table.
3. Please summarize what kind of inflammatory diseases are the different particle sizes used in PLGA-NPs? What is the principle?
4. Exosomes are the well-studied drug delivery targeting systems. Please compare the advantages and disadvantages of exosomes and PLGA-NPs delivery systems.
5. What is the loading efficiency and stability of PLGA-NPs compared to other drug carriers?
6. Please summarize the advantages and disadvantages of PLGA-NPs as a widely used carrier, and the future development direction.
The language of the article may be better polished
Author Response
REVIEWER 2
As a drug carrier with high tissue compatibility and strong operability, PLGA-NPs has been widely used. This review summarizes the application of PLGA-NPs in some inflammatory diseases and illustrates the principles of its possible role. However, there are still some problems that need to be solved.
- The article lists IBD, skin injury diseases, etc., but it does not cover some other common inflammatory diseases, such as pneumonia, asthma, etc. Can PLGA-NPs be applied to these diseases, or is it not reported? The authors had better summarize a table that can clearly show which diseases the PLGA-NPs delivery system has been applied.
We agree with the referee that PLGA-NPs have been extensively used to deliver immunosuppressive treatments for autoimmune and allergic diseases, or to elicit protective immune responses, such as in vaccination against infectious agents and cancer immunotherapy, but these aspects have been exhaustively treated in a recent review published in Pharmaceutics (ref 31), as described at the end of the Introduction. These applications are now highlighted and summarized in the new table (Table 1). To avoid overlapping with ref 31, we decided to focus on immune pathologic conditions not treated in that review, with a particular focus on the use of PLGA NPs in preclinical in vivo models of other diseases in which a key role is played by chronic inflammation or unbalance between the protective and the reparative phases of inflammation. The novel Table 2 summarizes the uses of PLGA NPs to support tissue repair described in our review
- Does the targeted delivery of PLGA-NPs mainly rely on particle size? Are there any modifications that can increase their organ-specific targeting? If so, please summarize the different modification methods and the principles in the table.
The description of several aspects of nanoparticle design has been extended in the Introduction section of the manuscript
This specific interaction with immune cells depends primarily on the size of the NPs, but this is not the only factor involved. Also surface properties, such as charge, hydrophobicity, and chemical modification, deeply affect the fate of the NPs [41]. The conjugation of targeting agents on the surface of NPs, for the specific recognition of the surface of target cells, is a well-established strategy to improve the cell-specific uptake of the nanomedicines. The most commonly used targeting agents are antibodies and low molecular weight ligands for cellular receptors. The specific molecules are selected according to the physio-pathological characteristics of the target cells [42]. For instance, glycosaminoglycans, such as hyaluronic acid and chondroitin sulfate, that bind selectively to CD44 receptors overexpressed by macrophages, is commonly used in inflammatory bowel diseases [43, 44]. Alternatively, ICAM1 and MAdCAM1, ligands of α4β7 integrin, were shown to increase the accumulation of NPs in a murine model of acute colitis [45]. Integrins are a good target to increase the site-specific accumulation of NPs at the inflammation site. A cyclic RGD peptide (arginine-glycine-aspartic acid) targeting αvβ3 integrins expressed on angiogenic vascular endothelial cells improved the accumulation of NPs in a Rheumatoid Arthritis model [46].
- Please summarize what kind of inflammatory diseases are the different particle sizes used in PLGA-NPs? What is the principle?
The description of several aspects of nanoparticle design has been extended in the Introduction section of the manuscript
Furthermore, PLGA-NPs, like many other colloidal nanocarriers, are easily taken up by the immune system, allowing for the selective delivery of anti-inflammatory drugs [39]. In general, NPs between 10 and 200 nm are commonly used as drug delivery systems, with a size-dependent toxicity (smaller particles are more toxic). NPs with an average size below 200 nm are typically internalized by dendritic cells, while NPs above 500 nm are preferentially captured by macrophages [40]. This specific interaction with immune cells depends primarily on the size of the NPs, but this is not the only factor involved. Also surface properties, such as charge, hydrophobicity, and chemical modification, deeply affect the fate of the NPs [41].
- Exosomes are the well-studied drug delivery targeting systems. Please compare the advantages and disadvantages of exosomes and PLGA-NPs delivery systems.
Exosomes are indeed emerging systems for the treatment of inflammatory disease. A brief comparison has been added to the text
PLGA-NPS are not the only colloidal systems exploited for the treatment of inflammatory diseases. Lipid NPs [50], gold NPs [51], and exosomes [52] are common alternatives to polymeric systems. Each nano-sized system exploits its own features to achieve an enhanced anti-inflammatory effect. Among them, exosomes are a particularly interesting, since they have an intrinsic immunomodulatory activity, depending on the cells from which they originate. The number of studies involving exosomes is rising, but, also for this technology, manufacturing issues are a limiting factor to clinical applicability [52].
- What is the loading efficiency and stability of PLGA-NPs compared to other drug carriers?
Comments on the encapsulation efficiency and stability were added when discussing the advantages of the formulation
Despite their hydrolyzable nature, they have an excellent shelf-life and can improve the stability of drugs in physiological fluids [47]. The encapsulation efficiency (EE%) for this type of formulation depends largely on the physicochemical properties of the drug, and can be modulated with chemical modification of the polymeric structure or different preparation techniques [23, 24].
- Please summarize the advantages and disadvantages of PLGA-NPs as a widely used carrier, and the future development direction.
Advantages and drawbacks are discussed in the following paragraphs
PLGA-NPs are particularly promising carriers to treat inflammatory diseases, due to the intrinsic features of the polymer, such as biocompatibility and biodegradability. Despite their hydrolyzable nature, they have an excellent shelf-life and can improve the stability of drugs in physiological fluids [47]. The encapsulation efficiency (EE%) for this type of formulation depends largely on the physicochemical properties of the drug, and can be modulated with chemical modification of the polymeric structure or different preparation techniques [23, 24].
The main drawbacks of this type of formulation are drug loading (typically around 1% w/w) and the high burst release of the loaded cargo. Furthermore, high production costs, and difficulties with large-scale production can limit the PLGA-NPs applicability in clinical practice [48]. Indeed, great efforts have been made at the industrial level to promote scale-up and improve PLGA-NPs manufacturing [49].
Comments on the Quality of English Language
The language of the article may be better polished
The English form has been revised
Reviewer 3 Report
Chronic inflammation contributes to diverse diseases, but conventional anti-inflammatory drugs have limited efficacy and adverse effects. Encapsulating bioactive molecules in nanoparticles, specifically poly (lactic-co-glycolic acid) (PLGA) NPs, enhances their properties. This review explores PLGA NPs' potential in preclinical models of diseases associated with chronic inflammation, emphasizing intestinal bowel disease, cardiovascular, neurodegenerative, osteoarticular, ocular diseases, and wound healing.
The review provides a substantial and comprehensive contribution, accurately reflecting the state of the art. The only aspect that needs improvement is a detailed description of how materials and processes can impact different nanoparticles and their applications. The review appears to focus too much on pharmacological aspects/diseases and less on nanoparticle design, which is the scope of the journal.
Author Response
Chronic inflammation contributes to diverse diseases, but conventional anti-inflammatory drugs have limited efficacy and adverse effects. Encapsulating bioactive molecules in nanoparticles, specifically poly (lactic-co-glycolic acid) (PLGA) NPs, enhances their properties. This review explores PLGA NPs' potential in preclinical models of diseases associated with chronic inflammation, emphasizing intestinal bowel disease, cardiovascular, neurodegenerative, osteoarticular, ocular diseases, and wound healing.
The review provides a substantial and comprehensive contribution, accurately reflecting the state of the art. The only aspect that needs improvement is a detailed description of how materials and processes can impact different nanoparticles and their applications. The review appears to focus too much on pharmacological aspects/diseases and less on nanoparticle design, which is the scope of the journal.
The description of several aspects of nanoparticle design has been extended in the Introduction section of the manuscript
Round 2
Reviewer 1 Report
The author optimized the paper and corrected some errors, I don't have any more opinions.
Easy to read and understand.
Reviewer 2 Report
The quality of the article has been greatly improved, and I have no further comments.
Quality of English language is good enough now.